# Prevalence and correlates of symptoms of depression, anxiety, and psychological distress among women of reproductive age with delayed conception in urban and peri-urban low to mid-socioeconomic neighborhoods of Delhi, India: A cross-sectional study

**Barsha Gadapani Pathak**[1,2], **Gitau Mburu**[3], **Ndema Habib**[3], **Rita Kabra**[3], **Aiysha Malik**[4], **James Kiarie**[3], **Ranadip Chowdhury**[1], **Neeta Dhabhai**[1], **Sarmila Mazumder**[1]*

1 Society for Applied Studies, New Delhi, India, 2 Centre for International Health, Faculty of Medicine, University of Bergen, Bergen, Norway, 3 UNDP-UNFPA-UNICEF-WHO-World Bank Special Program of Research, Development and Research Training in Human Reproduction (HRP) Department of Sexual and Reproductive Health and Research, World Health Organization, Geneva, Switzerland, 4 Department of Mental Health and Substance Use, World Health Organization, Geneva, Switzerland

* sarmila.mazumder@sas.org.in

## Abstract

## Introduction

One in six people of reproductive age experience infertility in their lifetime. Infertility can have significant impacts on mental health. Psychological distress is a broad term encompassing emotional suffering and mental health discomfort that can include symptoms of anxiety and depression but is not limited to these conditions. We investigated the prevalence of symptoms of depression, anxiety, and psychological distress and their associated risk factors among women of reproductive age with delayed conception.

## Methods

A total of 1530 women were recruited from community settings in Northern India. Quantitative data were collected using a 4-item Patient Health Questionnaire-4 (PHQ-4) which is an ultra-brief self-report questionnaire consisting of a 2-item depression scale (PHQ-2) and a 2-item anxiety scale (generalized anxiety depression-2). Data were collected between July 2020 and August 2021. Descriptive analysis was conducted to summarise the characteristics and prevalence of symptoms of depression, anxiety, and psychological distress. Logistic regression was used to identify risk factors for psychological distress.

**Data Availability Statement:** The dataset used for this study is derived from a larger dataset collected for multiple research objectives. Only the data relevant to the specific research question addressed in this study has been extracted and analyzed. Due to ethical restrictions and confidentiality agreements, the full dataset is not publicly available. However, access to the data may be requested by contacting the Ethics Review Committee (ERC) at the Society for Applied Studies via email at ERC@sas.org.in or by phone at +91-7838350052. The ERC will review data access requests and provide guidance regarding any applicable restrictions.

**Funding:** This work was supported by the UNDP-UNFPA-UNICEF-WHO-World Bank Special Programme of Research, Development, and Research Training in Human Reproduction (HRP), a co-sponsored program executed by the World Health Organization (WHO; https://www.who.int/) under TSA Grant Reference Number 2020/1026382-0. The funders had no role in study design, data collection and analysis, decision to publish, or preparation of the manuscript.

**Competing interests:** The authors have declared that no competing interests exist.

## Results

We obtained responses from all 1,530 women using these scales. Over half (54.31%) of participants had psychological distress, of whom 38.10% were experiencing mild distress, 10.59% moderate distress, and 5.62% severe distress. Additionally, 16.07% of participants reported symptoms of anxiety and 20% reported symptoms of depression. Factors associated with psychological distress were: (i) a higher total number of children that women intended to have in their lifetime, (ii) longer duration of trying to get pregnant (> 18 months), (iii) continuous effort trying to achieve pregnancy, (iv) women's perception that conception is taking long, (v) social isolation, (vi) being emotionally or verbally abused by husband and family members, and (vii) having other co-morbidities such as hypertension, irregular menstrual cycles, irregular bleeding between regular cycles, abnormal vaginal discharge or pain during sex.

## Conclusion

This study demonstrates the high mental health burden faced by women with delayed conception in low- to mid-socioeconomic neighbourhoods of North India including high levels of anxiety and depression. To better support individuals and couples achieve their reproductive goals, we advocate for the integration of psychosocial interventions to improve mental health outcomes and promote the well-being of those facing delays in achieving pregnancy. Specifically, addressing social isolation, fostering supportive networks, combating violence towards women, and incorporating fertility counselling and group-based psychosocial interventions within community and healthcare settings are needed to alleviate mental health symptoms among women who have difficulties in conceiving. However, the successful implementation of these recommendations may be challenged by the availability of the state's healthcare resources, necessitating tailored strategies with contextual adaptations.

## Introduction

Delays or difficulties to conceive or attain reproductive goals is a complex and multifaceted issue that affects millions of people [1, 2]. Although difficulties in achieving a pregnancy may be experienced for varying durations of time, the inability to conceive a clinical pregnancy after 12 months or more of regular unprotected sexual intercourse is defined as infertility [3]. Infertility is a pressing, yet often invisible public health problem globally [4]. In 2010, an estimated 48.5 million couples were reported to have infertility [5]. Currently, one in six people of reproductive age experience infertility in their lifetime [1]. In addition, lifetime prevalence of infertility does not differ much across the world, being 17.8% in high income counties and 16.5% in low and middle income countries [1].

Depression and anxiety are among the top ten leading causes of years lived with disability, globally [6]. Infertility or delayed conception can have significant impacts on mental health and wellbeing especially in cultural or social settings where having a child is highly valued [7–10]. In both men and women, infertility may contribute to poor mental health through several ways. First, receiving an infertility diagnosis can evoke feelings of guilt, shame, and fear of rejection [11–13]. Secondly, infertility is associated with stigmatization and humiliation within social contexts for both men and women [8], which can worsen mental health [14]. Third,

financial stress and uncertainties associated with treatment for infertility, including Assisted Reproductive Technology (ART) can contribute to distress for both men and women [15–20].

Among women, several factors may play a role to exacerbate the negative impact of infertility on their mental health. For instance, despite the contribution of male factors to difficulties in conceiving, women tend to face more severe social consequences and therefore are more likely to experience psychological distress [16, 21]. Sociocultural beliefs in many patriarchal societies perceive infertility as solely the woman's fault leading to infertile women's stigmatization [22, 23]. Moreover, women who have difficulties in conceiving are more likely to experience intimate partner violence [24, 25] and other relationship problems [8, 23], compared to women who achieve a pregnancy. Additionally, certain ethnic groups in LMICs, such as those from marginalized communities or lower socio-economic backgrounds, may report higher incidences of psychological distress due to compounded social pressures and limited access to healthcare [26, 27]. Studies have shown that women from Scheduled Castes and Scheduled Tribes often face greater challenges in accessing reproductive health services, which can exacerbate mental health issues in cases of delayed conception [28].

In India, infertility is widespread with reported prevalence of 12–14% [29, 30], yet there is insufficient focus on the mental health among women with infertility. Several researchers have highlighted this knowledge gap and the need to explore the prevalence, risk factors and sequelae of depression, anxiety, and psychological distress among Indian populations with infertility [31]. While anxiety and depression are specific mental health disorders characterized by distinct diagnostic criteria, psychological distress is a more general term that encompasses a range of emotional disturbances [32]. Anxiety and depression are often primary contributors to psychological distress, but the latter can also include stress, worry, and other forms of mental discomfort that may not necessarily fit into a specific psychiatric category [33, 34]. The need to address this gap is particularly relevant given the current government efforts to ensure that infertility services are in place and regulated across the country [35]. In this context, understanding the mental health situation among individuals and couples with infertility can inform current and future efforts to strengthen services for infertility. To respond to this gap, we conducted this cross-sectional study in Delhi, India to assess the burden of depression, anxiety and psychological distress and their associated risk factors among women of reproductive age with delayed conception.

## Methods

### Research question

This paper reports findings related to the question: What is the burden of mental health conditions and the associated factors among women who experience delay in conception?

### Study design

This is a cross-sectional study, and the above research question was one of several in a larger mixed methods study which researched the following three questions 1) What are the characteristics of women who experience delay in conception? 2) What is the quality of life and mental health of women who experience delay in conception? and 3) What were the experiences and actions taken among women who experience delay in conception? The protocol for this study is published elsewhere [36], and qualitative manuscripts related to the third question are in preparation [37, 38]. In this manuscript, we focus on both the first, and mental health component of the second question, and report on the prevalence of depression, anxiety and psychological distress symptoms among these women. This information will inform the development of interventions to improve mental health outcomes and promote well-being of

this population. Reporting of this study follows the STROBE guidelines for observational studies [39]. The minimum sample size needed to estimate infertility with 2% precision and 95% CI from this sample would be 1223, assuming a prevalence of 17% as reported in the literature, and a finite population of 12,500 [30, 40]. An allowance of 25% for rejection/ incomplete non-responses was made and the final sample size was 1530.

## Survey instrument

To assess mental health of women who did not conceive after 18 months, quantitative data were collected using a 4-item Patient Health Questionnaire-4 (PHQ-4). PHQ-4 is an ultra-brief self-report questionnaire that consists of a 2-item depression scale (PHQ-2) and a 2-item anxiety scale (GAD-2) [41], and it is a widely used instrument for screening for probable depression and anxiety. The combined scores of these two are then used to determine psychological distress. Numerous studies across various settings have demonstrated the PHQ-4 scale's as well as PHQ-2 scales' good internal consistency, reporting a Cronbach's alpha of 0.80 [42, 43]. Similarly, the GAD-4 (and shorter version GAD-2) scale has shown a Cronbach's alpha ranging from 0.75 to 0.81, indicating good reliability [44, 45].

## Study setting and population

The study recruited women of reproductive age who had participated in a randomized clinical trial known as Women and Infants Integrated Interventions for Growth Study (WINGS) in an urban low to mid-socioeconomic neighborhood located in the Delhi (which is in the Northern part of India) as detailed in the study protocol [36]. These neighborhoods were chosen based on the potential for demonstrating impact of interventions provided before and during pregnancy, based on evidence that poor health prior to conception (pre- and peri-conception period) is linked to birth outcomes [46]. Therefore, WINGS aimed to improve the health of women of reproductive age through a package of interventions related to nutrition, health, WASH, and psychosocial care to help them enter pregnancy in good health, free of sexually transmitted infections, and well-nourished [47]. At the end of 18 months those women who had not conceived were approached to participate in this infertility study. Therefore, participants in this infertility study were women at risk of becoming pregnant, sexually active, not using contraception, and not lactating, who reported trying unsuccessfully for pregnancy for 18 months.

## Recruitment, informed consent procedures and sample size

The recruitment for this study was conducted separately from the WINGS trial, targeting women who had completed 18 months of follow-up without achieving pregnancy. Any women who were > 49 years and currently pregnant were not considered eligible for this study. Consent was obtained at the time of their exit from WINGS, and research assistants contacted these women at least 14 days later to introduce the new study. Participants were consecutively recruited based on their exit dates, ensuring those who met the criteria were included. The study aimed for a sample size of 1,530, with recruitment continuing until this target was reached, ensuring a representative sample for assessing mental health outcomes. Women who were interested to participate were given an appointment during which detailed information about the study was provided by research assistants and informed consent obtained. The research assistants, who are nurses by training, checked whether the women understood the purpose of the study, its benefits, and risks. They provided women with detailed information about the study's aims and potential benefits prior to their enrollment. To ensure that women were not under duress to consent, clear statements were provided

explaining to the women that their participation was not obligatory but entirely voluntary. Written consent was obtained from all participants, who were reassured that their personal information would be kept confidential throughout the study. Once informed written consent was obtained, face-to-face response to the survey questionnaire was conducted. A total of 1530 women were recruited for this study. To minimize personal bias during face-to-face data collection, interviewers were trained to use neutral language and standardized protocols. Additionally, privacy was ensured during interviews, and participants were reassured about the confidentiality of their responses to reduce social desirability bias. For a more comprehensive and detailed description, please refer to the published protocol of this study [36].

## Outcomes

We used the Patient Health Questionnaire-4 (PHQ-4) to evaluate probable depression, anxiety, and psychological distress among study participants. The PHQ-4 is a validated 4-item self-report questionnaire that includes a 2-item depression scale (PHQ-2) and a 2-item anxiety scale (GAD-2) [41]. The four items are as follows, where items 1 and 2 assess depression (depression sub-scale) and items 3 and 4 assess anxiety (anxiety sub-scale): **1**. Over the last 2 weeks, how often have you had "little interest or pleasure in doing things"? **2.** Over the last 2 weeks, how often have you been "feeling down, depressed, or hopeless"? **3**. Over the last 2 weeks, how often have you felt "feeling nervous, anxious or on edge?" **4**. Over the last 2 weeks, how often have you "Not been able to stop or control worrying?

Scores on each subscale range from 0–6, with a score of 3 or greater considered positive for screening purposes, indicating probable depression and anxiety respectively. The total PHQ-4 scores range from 0–12, with a total score of psychological distress ranging from no distress to severe distress, as follows: None (0–2), Mild (3–5), Moderate (6–8), and Severe (9–12) [41]. Primary outcomes were therefore: i) Probable depression (score ≥3 or more), ii) Probable anxiety (score ≥3 or more) and iii) Overall probable psychological distress (mild, moderate, and severe).

## Operational definitions and outcome thresholds

**Distress** was assessed by PHQ 4 scales where scores from 0–2 indicate no distress 3–5 are categorized as mild, 6–8 as moderate and 9–12 as severe distress.

**Depression and Anxiety:** The PHQ 4 scale's sum of questions 1 and 2 (PHQ-2) are the sub-scale scores that determine depression (ranges from 0 to 6) and sum of questions 3 and 4 (GAD-2) determine the anxiety (score range, 0 to 6). On each subscale, a score of 3 or greater is considered positive symptoms for screening purposes.

## Exposure variables

The exposure variables were collected through questionnaires that have been adopted from previous surveys and studies related to infertility. The exposure variables included social demographic characteristics of the women and their families (such as age, education, occupation, religion, duration of marriage, wealth index, fertility intentions, substance use, and medical, and sexual history).

## Data analysis

The data were assessed for completeness, which involved examining missing values and generating summary statistics. Subsequently, the data were cleaned and analyzed using STATA version 16.0 (Stata Corp, Texas, USA). First, sample characteristics were summarized and for

descriptive statistics, continuous data were reported as mean and its standard deviation, and categorical variables as proportions.

The wealth index was determined through a principal component analysis, a method commonly employed in national surveys [48]. This analysis considered various household assets, including drinking water source, electricity source, sanitation facility type, cooking fuel type, and house construction materials. Ownership of items such as mattresses, pressure cookers, chairs, beds, tables, electronic devices, and vehicles was also considered. Additionally, factors like the number of household members per room and ownership of bank or post-office accounts contributed to the calculation [29]. The resulting total scores were used to categorize the population into five wealth quintiles: the poorest, very poor, poor, less poor, and the least poor [48, 49].

Subsequently logistic regression was performed to assess the factors (socio-demographic variables, fertility intentions, medical, sexual history, and addiction habits of women) associated with mental health status (distress or not distress) among women with delayed conception. We grouped quintiles, such as "least poor" and "less poor" into the first category, "poor" as the second, and "very poor" and "poorest" categorized as the third. This was done to simplify the socioeconomic classification system and facilitate a more straightforward and interpretable analysis [50]. The multivariable regression model was built following the stepwise backward method. The final multivariable regression model consisted of variables that were statistically significant (p-value<0.05) and the model was evaluated for independence of observations, specification error, goodness-of-fit, multicollinearity, and influential observations [51].

### Ethics approval

The study protocol was reviewed and approved by the ethics committee of Society for Applied Studies (SAS) (Approval number- SAS/ERC/RHR-Infertility/2020, approval date- January 9,2020) and the WHO Ethical Review Committee (WHO ERC) (Approval number- A-ID: A65998, approval date: February 13, 2020).

### Results

The baseline characteristics of the 1530 women included in this study are shown in **Table 1**.

The mean (SD) age of the women is 26.79 (3.28) years and the mean education of the women was 10.16 (4.27) years and only 5.95% (91/1530) of women were employed and the rest were housewives. Approximately 48.76% of the women (746/1530) had at least one living child and the mean (SD) of the age of the living child was 47.05 (29.72) months. Nearly 80% (1632/1985) were Hindu by religion, and nearly 40% (887/1985) were in the bottom two quintiles of the wealth index. The mean (SD) and median (IQR) family income per year was 2,801.21 (1,405.47) USD and 2,400.09 (1800.07 to 3360.13) USD respectively. Around 90% of the households had bank accounts, only 11.76% of the households had health scheme/health insurance and all enrolled households had concrete roofs and toilets, water connections within the house premises and legal electricity connections.

The general history and fertility intentions of the women who participated in the study are summarized in **Table 2**. The proportion of married women in the study was 99.93% (1529/1530) and 3.07% women adopted, fostered, or had stepchildren. The mean age of women at birth of 1st child was 22.15 (2.71) years, mean years a woman had been married to her current husband was 6.42 (3.23) years, and the duration of trying to conceive among women is 3.21 (2.08) years. More than 50% of the women had unprotected sexual intercourse more than twice in a week and 89.80% (1374/1530) women perceived delay in pregnancy and 49.87% (763/1530) thought that their husband perceived delay in pregnancy.

**Table 1. Baseline characteristics of the women included in the study.**

| Characteristic | (n, %) |
|---|---|
| **Total sample size** | **1530** |
| Women's Age (Mean/SD) | 26.79(3.28) |
| **Women with at least 1 living child** | 746 (48.76) |
| Mean (SD) months age of the child (N = 746) | 47.05 (29.72) |
| Family Structure: Proportion living in | |
| Nuclear | 661 (43.20) |
| Extended or Joint | 869 (56.80) |
| **Number of family members** | |
| Mean (SD) | 4.83 (2.74) |
| **Women's years of schooling (years)** | |
| Mean (SD) | 10.16 (4.27) |
| Never been to school | 80 (5.23) |
| **Women's current occupation** | |
| Employed$ | 91 (5.95) |
| Housewife | 1439 (94.05) |
| **Marital status** | |
| **Proportion of married women** | 1529 (99.93) |
| **The proportion of widowed women** | 1 (0.07) |
| **Husband's age in years** | |
| Mean (SD) | 28.62 (3.86) |
| **Husband's years of schooling** | |
| Mean (SD) | 10.99 (3.79) |
| Never been to school | 43 (2.81) |
| **Husband's current occupation** | |
| Government service | 14 (0.92) |
| Private job | 1094 (71.50) |
| Daily wager/ labourer | 68 (4.44) |
| Self employed | 290 (19.95) |
| Does not work | 64 (4.18) |
| **Household information** | |
| **Religion of the head of the household** | |
| Hindu | 1268 (82.88) |
| Muslim | 238 (15.56) |
| Christian | 7 (0.46) |
| Others* | 17(1.11) |
| **Proportion of members of the household owning this house or any other house** | 1171 (76.54) |
| **Proportion of the following members of the household owning this house or any other house** | **n = 1171** |
| Male | 965 (82.41) |
| Female | 199 (16.99) |
| Joint | 7 (0.60) |
| **Proportion of members of the household having a bank account or a post office account** | 1384 (90.46) |
| **Proportion of household having a Below poverty line card** | **73** (4.77) |
| **Proportion of members of the household having health scheme or health insurance** | 180 (11.76) |
| **Annual family income (in USD)** | |
| Mean (SD) | 2,801.21 (1,405.47) |
| **Socioeconomic and demographic details** | **n = 1530** |

*(Continued)*

**Table 1.** (Continued)

| Characteristic | (n, %) |
|---|---|
| **Wealth quintiles** | |
| Poorest | 306 (20.0) |
| Very poor | 306 (20.0) |
| Poor | 309 (20.20) |
| Less poor | 303 (19.80) |
| Least poor | 306 (20.0) |

$^\$$Government service, private job, daily wage/laborer, self-employed

*Other religions: Jain and Sikh.

The PHQ-4 scores among women are illustrated in **S1 Table** and we obtained responses from all 1,530 women using these scales. The findings include little interest or pleasure in activities being reported by over 50% (767/1530), and feeling down, depressed, or hopeless was

**Table 2. Reproductive history and fertility intentions of the women.**

| Variables | N = 1530 Number (%) |
|---|---|
| **The proportion of women who never had a child** | 755 (49.35) |
| **Women who have any adopted, fostered, or stepchildren** | 47 (3.07) |
| **Maternal age in years at first birth** (Mean, SD) | 22.15 (2.71) |
| **Years woman has been married with her current husband** (Mean, SD) | 6.42 (3.23) |
| **Partner fathered a child** (from another woman) | 626 (40.92) |
| **Previous pregnancy intention** [before enrolment in the primary trial*] | 1224 (80.0) |
| **Duration of trial to conceive** (in years) (Mean, SD) | 3.21 (2.08) |
| **Unprotected sexual intercourse** | |
| Once every month | 27 (1.76) |
| Twice a month | 100 (6.54) |
| Once a week | 199 (13.01) |
| Twice a week | 433 (28.30) |
| More than twice a week | 771 (50.39) |
| **Delay in pregnancy** (as per Woman) | 1374 (89.80) |
| **Delay in pregnancy** (as per husband) | 763 (49.87) |
| **Experienced from partner** | |
| Physical abuse | 62 (4.05) |
| Emotional abuse | 262 (17.12) |
| Verbal abuse | 131 (8.56) |
| Denial of financial support | 34 (2.22) |
| Divorce | 30 (1.96) |
| Abandonment | 31 (2.03) |
| **Experienced from anyone else in the family for taking too long to become pregnant** | |
| Physical abuse | 47 (3.07) |
| Emotional abuse | 748 (48.89) |
| Verbal abuse | 279 (18.24) |
| Denial of financial support | 51 (3.33) |
| Divorce | 41 (2.68) |
| Abandonment | 51 (3.33) |

*The participants of this study were earlier a part of trial names as WINGs.

**Table 3. Prevalence of probably/symptoms of depression, anxiety, and distress among the participants.**

| Mental health status (N = 1530) | n (%) |
|---|---|
| Mental distress (PHQ4 Scale) | |
| Not Distressed (0 to ≤2) | 699 (45.69%) |
| Distressed (≥3 to ≤12) | 831 (54.31%) |
| Mild distressed (≥3 and≤5) | 583 (38.10%) |
| Moderate distress (≥6 and ≤8) | 162 (10.59%) |
| Severe distress (≥9 and ≤12) | 86(5.62%) |
| Anxiety (GAD-2 subscale) | |
| No Anxiety (<3) | 1284(83.92) |
| Anxiety (≥ 3) | 246 (16.07) |
| Depression (PHQ-2 subscale | |
| No depression (<3) | 1224(80.00%) |
| Depression (≥ 3 | 306 (20.00%) |

noted by more than 50% (779/1530) for several days over last 2 weeks. Additionally, 39.48% (604/1530) experienced nervousness, anxiety, or unease, and 35.62% (545/1530) struggled to stop or control worrying for several days for the past 2 weeks. The mean (SD) and median (IQR) PHQ 4 scores were 3.20 (2.90) 3(0–4) respectively.

The mental health of the 1530 women who participated in the study is summarized in **Table 3**. Results from the PHQ-4 depression sub-scale revealed that 54.31% (831/1530) of individuals fell into the distressed category, with 38.10% (583/1530) experiencing mild distress, 10.59% (162/1530) reporting moderate distress, and 5.62% (86/1530) indicating severe distress. Furthermore, the GAD-2 subscale, focusing on anxiety, highlighted that 16.07% (246/1530) of the participants exhibited symptoms of anxiety and the PHQ-2 subscale, targeting depression, unveiled that 20% (306/1530) of the individuals had signs of depression.

In the multivariable logistic regression, psychological distress was significantly associated with several variables (**Table 4**). Specifically, higher number of intended total children by the women significantly increased distress (AOR: 2.23; 95% CI: 1.68–3.13), women trying to conceive >18 months significantly increased distress (AOR: 2.22; 95% CI: 1.75–3.12), women were at 2.54 times (AOR: 2.54; 95% CI: 1.64–3.94) increased odds of distress who were continuously making an effort for pregnancy since enrolment in the study, women who perceived taking longer time for conception were significantly at increased distress (AOR: 2.23; 95% CI: 1.43–3.49). Additionally, women experiencing social isolation had 1.81 times higher odds of distress (AOR: 1.80; 95% CI: 1.29–2.36), those that were verbally abused family members had 6.88 times higher odds (AOR: 6.88; 95% CI: 4.36–10.87) those that were emotionally abused family members had 1.62 times higher odds (AOR 1.62; 95% CI: 1.07–2.24) and those that were emotionally abused by husbands had 1.51 times higher odds (AOR: 1.51; 95% CI: 1.02–2.31) of experiencing distress. Medical conditions, specifically irregular menstrual cycles, bleeding between the regular menstrual cycles and hypertension increased the odds of distress in women by 3.2 (AOR: 3.20; CI: 4.36–10.87), 1.97 (AOR: 1.98; CI: 1.04–3.67) and 5.36 times (AOR: 3.20; 95% CI: 4.36–10.87) respectively. Abnormal vaginal discharge increased odds of distress by 1.48 (AOR: 1.48; 95%CI: 1.11–1.97), and pain during sexual intercourse increased odds of distress by 1.66 (AOR: 1.64; 95% CI: 1.22–2.21).

## Discussion

This cross-sectional study investigated the prevalence and risk factors associated with probable depression, anxiety, and psychological distress (assessed by PHQ-4) among women of

**Table 4. Findings from univariable and multivariable logistic regression for factors determining distress among women with delayed conception.**

| Variables | Distressed women | |
| --- | --- | --- |
| | Unadjusted OR (95% CI), (p-value) | Adjusted OR (95% CI), (p-value) |
| **PARTNERSHIP AND CHILDREN'S CIRCUMSTANCES** | | |
| Women who had at least one [living] child<br>Yes<br>No | 0.68* (0.55,0.84) (p = 0.01)<br>Reference | |
| Total number of children that women intend to have in their lifetime. ¶ | 1.87**(1.43, 2.44) | 2.23** (1.68,3.13) ((p = <0.001***) |
| Women age at birth of 1st child | | |
| 18 to 21 years | Reference | |
| 22 to 27 years | 1.17* (0.87, 1.57) (p = 0.20) | |
| 28 to 35 years | 0.65 (.028, 1.54) (p = 0.34) | |
| Husband's age | 0.96* (0.94,0.99) (p = 0.02) | |
| Husband's education | 0.89*(0.93,0.99) (p = 0.04) | |
| **FERTILITY INTENTIONS** | | |
| Women trying to get pregnant over 18 months (i.e., prior to joining the previous trial).<br>Yes<br>No | 3.98** (3.02,5.23)<br>Reference | 2.22**(1.75,3.12)<br>Reference |
| Husband fathered other children.<br>Yes<br>No | 0.68**(0.55, 0.83)<br>Reference | |
| Total duration trying to conceive | 1.15** (1.09,1.21) | |
| Throughout the WINGS trial, there was a continuous effort to achieve pregnancy<br>Yes<br>No | 2.65** (1.83,3.84)<br>Reference | 2.54**(1.64,3.93) ((p = <0.001***)<br>Reference |
| Perception of the woman that conception is taking longer.<br>Yes<br>No | 3.41** (2.37,4.91)<br>Reference | 2.23** (1.43,3.49)<br>Reference |
| Perception of the husband/partner that conception is taking longer.<br>Yes<br>No | 1.41* (1.15,1.72)<br>Reference | |
| Women felt isolated.<br>Yes<br>No | 2.77** (2.2,3.43)<br>Reference | 1.81** (1.29,2.36)<br>Reference |
| Women physically abused by partner/husband.<br>Yes<br>No | 9.30** (5.46,15.83)<br>Reference | |
| Women emotionally abused by partner/husband.<br>Yes<br>No | 8.36** (6.15,11.36)<br>Reference | 1.51* (1.02,2.31)<br>Reference |
| Women verbally abused by their partner/husband.<br>Yes<br>No | 11.74** (7.96,17.31)<br>Reference | |
| Women divorced by partner/husband.<br>Yes<br>No | 5.88* (2.04,16.96)<br>Reference | |
| Women denied financial support by their partner/husband.<br>Yes<br>No | 9.44** (2.87,31.01)<br>Reference | |

*(Continued)*

**Table 4.** (Continued)

| Variables | Distressed women | |
|---|---|---|
| | **Unadjusted OR (95% CI), (p-value)** | **Adjusted OR (95% CI), (p-value)** |
| Women are physically abused by their family members. Yes No | 9.48** (3.38,26.55) Reference | |
| Women verbally abused by family members. Yes No | 11.87** (7.75,18.16) Reference | 6.88* (4.36,10.87) Reference |
| Women emotionally abused by family members. Yes No | 1.44** (1.18,1.77) Reference | 1.62* (1.07,2.24) Reference |
| **MEDICAL HISTORY** | | |
| Regular period every month Yes No | 0.56**(0.41,0.77) Reference | |
| Irregular menstrual cycles in women. Yes No | 3.33**(1.71,6.52) Reference | 3.22** (4.36,10.87) Reference |
| Bleeding between the regular menstrual cycles in women. Yes No | 2.47* (1.43,4.26) Reference | 1.98* (1.04,3.67) Reference |
| Women having painful periods. Yes No | 1.46* (1.17,1.84) Reference | |
| Women suffering from hypertension. Yes No | 3.95*(1.49,10.44) Reference | 5.36**(2.00,14.38) Reference |
| **SEXUAL HISTORY OF THE WOMEN** | | |
| Abnormal vaginal discharge Yes No | 1.83**(1.44,2.33) Reference | 1.48*(1.11,1.97) Reference |
| Women have pain during sexual intercourse. Yes No | 2.15**(1.68,2.77) Reference | 1.66*(1.22,2.21) Reference |
| Women have bleeding after sexual intercourse. Yes No | 2.44* (1.02,5.80) Reference | |
| Women who perceive she or her husband have a problem during intercourse. Yes No | 1.60**(1.23,2.07) Reference | |
| **ADDICTION IN WOMEN AND THEIR SPOUSES/PARTNERS** | | |
| Alcohol consumption by women Yes No | 3.75* (1.18,11.91) Reference | |

In the univariable analysis p-value ≤0.20 is considered significant, Multivariable analysis p-value ≤0.05 is considered significant

*p-value is any value from 0.01 to 0.05

** p-value is ≤0.001

¶Total number of children that women intend to have in their lifetime refers to the desired or planned number of children that a woman anticipates having over the course of her reproductive life. It reflects the woman's reproductive intentions or aspirations for family size.

reproductive age with delayed conception in low- to mid-socioeconomic neighbourhoods of Delhi, India. Over half (54.31%) of participants had symptoms of distress, among whom 38.10% had mild distress, 10.59% moderate distress, and 5.62% severe distress. Additionally,16.07% of participants showed symptoms of anxiety and 20% exhibited symptoms of depression.

The National Mental Health Survey of India 2016, which collected data from multiple sites through a nationwide household survey using a uniform methodology, reported a prevalence of common mental disorders, including depression and anxiety, of 5.8% among women of reproductive age (WRA) [52]. But several studies, globally and in India, comparing mental health outcomes between fertile and infertile women have consistently demonstrated a significantly higher burden of psychological distress among infertile women. One study reported that infertile women had a significantly higher mean depression score on the Beck Depression Inventory (11.84) compared to fertile women (4.06), with 15.7% of infertile women experiencing moderate depression, compared to none in the fertile group [53]. Another study found that infertile women had twice the prevalence of depression compared to controls, with those experiencing infertility for 2–3 years showing significantly higher depression scores than those with shorter or longer durations of infertility [54]. These comparisons underscore the greater psychological impact of infertility, which is further corroborated by the high prevalence of distress, anxiety, and depression in our study population. Furthermore, the high prevalence of anxiety (16.07%) and depression (20%) and psychological distress (44.3%) among women with delayed conception are generally consistent with findings from other studies from LMICs [17, 55, 56], although wide prevalence ranges have also been reported from these settings [57, 58]. Reviews that include women from high income countries have also found high prevalence of mental health problems including depression among women with infertility [58], underscoring a strong association between mental health and delays in pregnancy. An elevated PHQ–4 score is not diagnostic, but is, instead, an indicator for the need for further inquiry to establish the presence or absence of a clinical disorder warranting treatment.

Our study identified several factors that are independently associated with psychological distress, which have several implications on potential interventions. First, the duration of trying to conceive emerged as a significant predictor of distress among women with delayed conception, a finding that is similar to other studies [59]. This finding suggests that providing psychosocial support early in peoples fertility care journeys could have a beneficial effect. A practical way of achieving this is including psychosocial assessments as part of initial fertility care appointments, ensuring that healthcare providers listen to patients and discuss whether they need psychosocial and emotional support, and providing it or referring patients to it if they need it.

Second, the finding that a larger family size was associated with an increased risk of distress is not unexpected, given that the delay in conceiving means that women would have to wait even longer to achieve their reproductive goals. Other studies have shown that when fertility goals are not attained as desired, individuals become distressed [60]. Similarly, prolonged attempts to conceive was associated with higher risk of distress in our study, which supports the preposition by other scholars that unrealised fertility intent has an effect on distress [61]. This finding suggests that mental interventions would be particularly beneficial for women who are 'trying' to become pregnant as opposed to those who simply do not conceive despite having regular intercourse without contraception. To support reproductive autonomy, early identification and support for women who conceive later-than-desired, to ensure that they realize their desired pregnancy timing [62]. It is also notable that women who perceived that conception was taking too long were at an increased risk of distress, compared to those that did not have such perception, regardless of the actual time taken trying to conceive. This

might be related to the environments that these women were in, which as discussed below, included exposure to different forms of violence, and social isolation.

Third, emotional abuse by husbands, and both emotional and verbal abuse by family members were associated with distress, which is consistent with studies linking violence with poor mental health among other populations of women [63]. This finding indicates the need for community-based and multisectoral interventions to prevent violence among women with delayed conception, and to educate communities regarding the need to support, rather than blame these women. Studies have shown that violence against women with infertility is common [25], and spousal and family support is protective in mitigating distress [64, 65]. Awareness and reduction of violence at the community level is warranted given the high lifetime prevalence (35%) of violence towards ever-married/partnered women in India [66]. In addition to community-based interventions, healthcare providers have a role to play in ensuring women are made aware or referred to available services for intimate partners and other forms of family violence such as counselling, shelters, among others, in case they need them.

Fourth, our study also shows that social isolation was associated with distress, which reinforces the importance of social support, including peer-based support, that can provide opportunities for women to share their experience with others. Perceived social support has been shown to improve depressive symptoms in other studies [67]. In our study setting, there were no patient support organizations specifically for people struggling with conception, where our study participants could readily access peer-based support. In this context, facilitator-guided, group-based, self-help interventions that have been shown to mitigate distress among other populations [68], could be useful. To ensure that such groups are linked to other community health initiatives and are not implemented in isolation, Accredited Social Health Activist (ASHAs), who are community-based workers, could facilitate and enroll women. Participation in group psychological intervention has been shown to improve mental health and infertility-related stress among people with infertility [69]. Although effectiveness could vary based on context, brief psychological interventions delivered by non-specialists in community-settings of low resource settings lacking specialist mental health workforce could be an effective way of addressing mental health problems. From the health systems perspective, greater availability of counselling for fertility problems within the health services could be one way of ensuring that women do not feel isolated and have a reliable source of information about infertility problems.

Finally, our study identified an association between distress and physical comorbidities such irregular menstrual cycles, abnormal vaginal discharge, and hypertension. These findings are consistent with the literature among general populations showing association between depression and anxiety among patients with hypertension [70, 71], which may occur due to altered autonomic mechanisms [72]. Our findings also support reports from other studies showing that abnormal vaginal discharge or menstrual patterns can interfere with quality of life and be a source of distress among many women [73]. In women who are having difficulties in achieving a pregnancy, the effects of abnormal vaginal discharge or irregular menstrual patterns on mental health could potentially be mediated by their impact on sexual well-being, intimacy and general stress on relationship with male partners [74].

## Strengths of this study

This study focuses on a socially sensitive and understudied population of women facing delayed conception in a low-resource setting, highlighting a crucial yet often neglected aspect of sexual and reproductive health. We have conducted a comprehensive assessment by utilizing PHQ-4 and have separately reported the findings of subscales GAD-2, and PHQ-2, thereby

providing a multifaceted perspective on mental well-being, encompassing distress, anxiety, and depression. The study goes beyond traditional infertility-related factors to explore social, emotional, and physical components contributing to distress, offering a more holistic understanding of the mental health status of participants. Additionally, this study is conducted at community level compared to many others which are conducted within clinical settings.

## Study limitations

As a cross-sectional study, this research cannot establish causal relationships between identified risk factors and mental health. Additionally, we relied on self-reported data by women which may introduce social desirability bias to the findings. The study's confined setting, a single city in North India, limits generalizability of our findings to other regions, countries, and populations. Our study was conducted in an urban setting, which may differ from rural and tribal communities where access to fertility care is even lower. To enhance generalizability, comprehensive multi-site study that covers diverse regions across India would be required. Additionally, the study was conducted during the pandemic, which could have influenced the prevalence of anxiety and depression. While the pandemic might have heightened psychological distress, it also reflects the real-world challenges faced by women during this time, making our findings relevant to similar crisis situations. Our qualitative findings from focussed group discussions and in-depth interviews revealed that most women attributed their psychological distress primarily to issues related to delayed conception, with limited mention of the pandemic as a contributing factor [75]. Finally, it should be noted that our study does not provide diagnosis of anxiety or depression, since elevated PHQ–4 scores are not diagnostic per se, but instead, they do indicate the presence of symptoms and the potential need for further efforts to confirm diagnosis in individual patients, as a screening strategy [41]. Despite these limitations, our study provides useful information pointing to the need for mental health interventions for women facing difficulties in conceiving in the study community.

## Conclusion

This study sheds light on the significant mental health burden faced by women with delayed conception in low- to mid-socioeconomic neighbourhoods of Delhi in India and reveals high levels of distress, anxiety, and depression. Social isolation, abuse, and several comorbidities were associated with psychological distress. To effectively mitigate these mental health problems, a multi-pronged approach is needed, integrating psychosocial support in infertility prevention, diagnosis, and treatment. Evidence-based strategies, such as early screening for psychological distress using tools like the PHQ-4 and routine mental health assessments for women undergoing infertility treatment, can effectively identify those in need of support. Additionally, addressing social isolation, fostering supportive networks, combating violence, and incorporating fertility counseling, peer support groups, and mental health education into community and healthcare settings are crucial steps to alleviate the mental health burden among women facing difficulties in conceiving.

The successful implementation of these recommendations may face challenges based on the availability of financial, infrastructural, and human resources within the state's healthcare system. Therefore, it is necessary to have tailored strategies that ensure adequate resources are in place for effective intervention.

## Supporting information

**S1 Table. Mental health status of women assessed by PHQ-4.**
(DOCX)

## Acknowledgments

Authors thank Inka Weissbecker for commenting on an earlier draft. We would like to express our sincere gratitude to the entire team involved in the data collection, as well as our data management team, with special thanks to Baljeet Kaur for her contributions.

## Author Contributions

**Conceptualization:** Gitau Mburu, Rita Kabra, James Kiarie, Ranadip Chowdhury, Sarmila Mazumder.

**Data curation:** Sarmila Mazumder.

**Formal analysis:** Barsha Gadapani Pathak.

**Funding acquisition:** Sarmila Mazumder.

**Methodology:** Gitau Mburu, Ndema Habib, Rita Kabra, Aiysha Malik, Ranadip Chowdhury, Sarmila Mazumder.

**Project administration:** Sarmila Mazumder.

**Resources:** Gitau Mburu, Rita Kabra, James Kiarie.

**Supervision:** Gitau Mburu, Ndema Habib, Sarmila Mazumder.

**Validation:** Barsha Gadapani Pathak, Gitau Mburu, Ndema Habib, Sarmila Mazumder.

**Visualization:** Barsha Gadapani Pathak, Gitau Mburu.

**Writing – original draft:** Barsha Gadapani Pathak.

**Writing – review & editing:** Barsha Gadapani Pathak, Gitau Mburu, Ndema Habib, Rita Kabra, Aiysha Malik, James Kiarie, Ranadip Chowdhury, Neeta Dhabhai, Sarmila Mazumder.

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
