## [Decision Letter · Decision Letter 0]

29 Jul 2024

PONE-D-24-11366Prevalence and correlates of symptoms of depression, anxiety,and psychological distress among women of reproductive age with delayed conception in urban and peri-urban low to mid-socioeconomic neighborhoods of Delhi, India: A cross-sectional study.PLOS ONE

Dear Dr. Mazumder,

Thank you for submitting your manuscript to PLOS ONE. After careful consideration, we feel that it has merit but does not fully meet PLOS ONE’s publication criteria as it currently stands. Therefore, we invite you to submit a revised version of the manuscript that addresses the points raised during the review process.

We look forward to receiving your revised manuscript.

Kind regards,

Yuan-Pang Wang, M.D., Ph.D.

Academic Editor

PLOS ONE

“Financial Disclosure Statement

This work received funding from the UNDP-UNFPA-UNICEF-WHO-World Bank Special Programme of Research, Development, and Research Training in Human Reproduction (HRP), a co-sponsored program executed by the World Health Organization (WHO). TSA Grant Reference Number: 2020/1026382-0.

url:https://www.who.int/”

“This work received funding from the UNDP-UNFPA-UNICEF-WHO-World Bank Special Programme of Research, Development and Research Training in Human Reproduction (HRP), a co-sponsored programme executed by the World Health Organization (WHO). TSA Grant Reference Number: 2020/1026382-0.”

“Financial Disclosure Statement

This work received funding from the UNDP-UNFPA-UNICEF-WHO-World Bank Special Programme of Research, Development, and Research Training in Human Reproduction (HRP), a co-sponsored program executed by the World Health Organization (WHO). TSA Grant Reference Number: 2020/1026382-0.

url:https://www.who.int/”

4. In the online submission form, you indicated that your data is available only on request from a third party. Please note that your Data Availability Statement is currently missing the name of the third party contact or institution / contact details for the third party, such as an email address or a link to where data requests can be made. Please update your statement with the missing information.

Additional Editor Comments:

Both reviewers see contribution of this paper. Although we cannot accept the manuscript for publication by now, please point to point address reviewers' comments. I am happy to see a reworked submission.

Reviewers' comments:

Reviewer's Responses to Questions

**Comments to the Author**

1. Is the manuscript technically sound, and do the data support the conclusions?

Reviewer #1: Yes

Reviewer #2: Yes

2. Has the statistical analysis been performed appropriately and rigorously? 

Reviewer #1: Yes

Reviewer #2: I Don't Know

3. Have the authors made all data underlying the findings in their manuscript fully available?

Reviewer #1: Yes

Reviewer #2: Yes

4. Is the manuscript presented in an intelligible fashion and written in standard English?

Reviewer #1: Yes

Reviewer #2: Yes

5. Review Comments to the Author

Reviewer #1: The manuscript titled “Prevalence and correlates of symptoms of depression, anxiety, and psychological distress among women of reproductive age with delayed conception in urban and periurban low to mid-socioeconomic neighborhoods of Delhi, India: A cross-sectional study” by B. G. Pathak and co-workers have investigated the occurrence of mental health problems (including depression, anxiety, psychological distress) among infertile women. The manuscript is informative and the findings are of significant importance since there are rising number of infertility cases in India. The findings supported that mental health problems in women delayed pregnancy. However, the authors should also take into account the prevalence of psychological distress among the fertile women carried out by previous studies and compare their findings with the present study. The authors should also consider association of women’s mental health with determinants such as educational level, age, employment status, alcohol addiction problem of husband or other family members living with the subjects. In the methodology section, please mention how sample size was calculated and what sampling method was used. The strategies required to identify women at increased risk of mental disorders due to inability to conceive and approaches to provide support systems needs to be elaborated in the discussion part.

Reviewer #2: Manuscript Number: PONE-D-24-11366

Title: Prevalence and correlates of symptoms of depression, anxiety, and psychological

distress among women of reproductive age with delayed conception in urban and periurban

low to mid-socioeconomic neighborhoods of Delhi, India: A cross-sectional

study.

Thank you for giving me the opportunity to review the above-titled manuscript. I have the following comments about the manuscript:

Abstract

1.Please define psychological distress and how it differs from anxiety or depression.

2.What was the response rate?

3.Are there any factors that could interfere with implementing the recommendations? Does this state in India have enough resources to provide such recommended services? If not, the implementation should be subject to the availability of resources.

Introduction

4.Are there any particular ethnic groups in India that report a higher incidence of psychological distress such as depression or anxiety as the results of delayed conception?

5.What are the differences between depression and anxiety on one side and psychological distress on the other, given that anxiety and depression are the main causes of psychological distress? This needs more explanation and definitions within the context of the literature.

Methods

6.What was the sample size calculation?

7.What were the exclusion criteria? Were women who were diagnosed with anxiety and depression, either with or without treatment, excluded? How were they identified?

8.How was the face-to-face response during data collection eliminated or at least minimized to prevent personal bias? Did the authors address these issues?

9.Please provide details on how the women were recruited, even though this study is part of a larger referenced study.

10.What were the reliability measures (Cronbach's alpha) for each instrument used to collect data?

11.Emotional abuse by husbands can cause psychological distress (either anxiety or depression) and social isolation can be a result of psychological distress. How were these causative or resultant factors controlled during the analysis?

12.Were any regression analyses conducted according to ethnic/religious backgrounds? What were the results?

Study's Limitation

13.In addition to the limited generalizability of the study, as it has been conducted in a single city in North India, it was also conducted during the COVID-19 pandemic, which might have influenced the rates of anxiety or depression during this time. How was this accounted for? This might be a limitation of the study as people might have had a higher chance of developing psychological symptoms such as anxiety and depression during COVID-19 pandemic.

Conclusion:

14.Although there were many recommendations that emerged from this study, their implementation is subject to the availability of resources, including financial aspects. This needs to be highlighted.

6. PLOS authors have the option to publish the peer review history of their article (what does this mean?). If published, this will include your full peer review and any attached files.

Reviewer #1: No

Reviewer #2: No

---

## [Author Response · Author response to Decision Letter 0]

6 Sep 2024

AUTHOR RESPONSE LETTER

Dear

Editor-in-chief

PLOS ONE journal. 

We thank you for your response and invitation to resubmit the paper to the PLOS ONE journal. We have taken note of the comments and suggestions made by the reviewers and editor and have revised the manuscript accordingly.

We have included the author’s responses below the original reviewer’s and editor’s comments. Where revision to the manuscript text has been made, the text is copied into this letter with blue highlights in italics (as they appear in the new/clean version) and references are inserted too. 

Both this clean version of the revised manuscript, as well as a version with the changes tracked (highlighted), are also submitted.

Best regards,

Sarmila Mazumder 

Reviewer’s comments and Author’s responses 

Reviewer 1: 

Comment 1: The manuscript titled “Prevalence and correlates of symptoms of depression, anxiety, and psychological distress among women of reproductive age with delayed conception in urban and periurban low to mid-socioeconomic neighborhoods of Delhi, India: A cross-sectional study” by B. G. Pathak and co-workers have investigated the occurrence of mental health problems (including depression, anxiety, psychological distress) among infertile women. The manuscript is informative and the findings are of significant importance since there are rising number of infertility cases in India. The findings supported that mental health problems in women delayed pregnancy.

Response 1: Thank you for your positive feedback. We appreciate your recognition of our study findings, especially considering the rising infertility cases in India.

Comment 2: However, the authors should also take into account the prevalence of psychological distress among the fertile women carried out by previous studies and compare their findings with the present study. 

Response 2: Thank you for the excellent suggestion and feedback. We have incorporated a write-up in the discussion section incorporating your suggestion “The National Mental Health Survey of India 2016, which collected data from multiple sites through a nationwide household survey using a uniform methodology, reported a prevalence of common mental disorders, including depression and anxiety, of 5.8% among women of reproductive age (WRA).(Annajigowda et al. 2023)But several studies, globally and in India, comparing mental health outcomes between fertile and infertile women have consistently demonstrated a significantly higher burden of psychological distress among infertile women. One study reported that infertile women had a significantly higher mean depression score on the Beck Depression Inventory (11.84) compared to fertile women (4.06), with 15.7% of infertile women experiencing moderate depression, compared to none in the fertile group.(Verma and Baniya 2016) Another study found that infertile women had twice the prevalence of depression compared to controls, with those experiencing infertility for 2-3 years showing significantly higher depression scores than those with shorter or longer durations of infertility.(Domar et al. 1992)” 

Comment 3: The authors should also consider association of women’s mental health with determinants such as educational level, age, employment status, alcohol addiction problem of husband or other family members living with the subjects. 

Response 3: Thank you for the suggestion. We conducted univariable analyses considering educational level, age, employment status, and the husband's alcohol addiction problems. While the husband's age and education showed significant associations in the univariable regression analysis, no significant associations were found for the husband's alcohol consumption or occupation. However, in the multivariable model, the husband's age and education were no longer significant. We have added the significant findings from the univariable regression analysis to Table 4.

Comment 4: In the methodology section, please mention how sample size was calculated and what sampling method was used. 

Response 4: Thank you for the feedback. We have included the sentence in the methodology regarding the sample size “The minimum sample size needed to estimate infertility with 2% precision and 95% CI from this sample would be 1223, assuming a prevalence of 17% as reported in the literature, and a finite population of 12,500.(Rutstein 2016; Adamson et al. 2011) An allowance of 25% for rejection/ incomplete non-responses was made and the final sample size was 1530.” 

Additionally, the recruitment process and strategy for selection of participants in this study was carefully designed to ensure a systematic and unbiased selection of participants. Although this study is part of a larger referenced study (WINGS), the recruitment for the current study was conducted separately and followed a distinct protocol.

Participants were recruited from women who had completed 18 months of follow-up in the WINGS trial without achieving pregnancy. Consent to participate in this new study on failure to conceive was sought at the time of their exit from WINGS. These research assistants contacted the women at least 14 days after their exit from WINGS (primary trial) to introduce the new study. Women were recruited consecutively based on their exit dates from WINGS, ensuring that those who met the study criteria (failure to conceive after 18 months) were invited to participate. The sample size for this study was calculated to ensure adequate precision, with a target of 1,530 participants. Recruitment continued until this sample size was achieved, based on preliminary data suggesting that approximately 50% of the WINGS participants had not conceived by the end of the 18 months. This methodical approach ensured a robust and representative sample for assessing the mental health outcomes of interest.

We have added this strategy to the manuscript’s methods section “The recruitment for this study was conducted separately from the WINGS trial, targeting women who had completed 18 months of follow-up without achieving pregnancy. Consent was obtained at the time of their exit from WINGS, and research assistants contacted these women at least 14 days later to introduce the new study. Participants were consecutively recruited based on their exit dates, ensuring those who met the criteria were included. The study aimed for a sample size of 1,530, with recruitment continuing until this target was reached, ensuring a representative sample for assessing mental health outcomes.”

Comment 5: The strategies required to identify women at increased risk of mental disorders due to inability to conceive and approaches to provide support systems needs to be elaborated in the discussion part.

Response 5: Thank you for the valuable suggestion. We agree that elaborating on the strategies to identify women at increased risk of mental disorders due to infertility, as well as approaches to providing appropriate support systems, is crucial.

We have modified the conclusion section of the manuscript and incorporated certain strategies “To effectively mitigate these mental health problems, a multi-pronged approach is needed, integrating psychosocial support in infertility prevention, diagnosis, and treatment. Evidence-based strategies, such as early screening for psychological distress using tools like the PHQ-4 and routine mental health assessments for women undergoing infertility treatment, can effectively identify those in need of support. Additionally, addressing social isolation, fostering supportive networks, combating violence, and incorporating fertility counseling, peer support groups, and mental health education into community and healthcare settings are crucial steps to alleviate the mental health burden among women facing difficulties in conceiving."

Reviewer 2: 

Section: Abstract

Comment 1: Please define psychological distress and how it differs from anxiety or depression.

Response 1: Thank you for the feedback. We have added this sentence in the abstract “Psychological distress is a broad term encompassing emotional suffering and mental health discomfort that can include symptoms of anxiety and depression but is not limited to these conditions.” 

Comment 2: What was the response rate?

Response 2: Thank you for the query. We acquired responses from all 1530 women for the scales. We have updated the information in abstract and result-section “We obtained responses from all 1,530 women using these scales”

Comment 3: Are there any factors that could interfere with implementing the recommendations? Does this state in India have enough resources to provide such recommended services? If not, the implementation should be subject to the availability of resources.

Response 3: Thank you for the suggestions and feedback. We understand that there are financial, human resources and infrastructure related issues, which may hinder recommendation implementation. We have modified and added a write-up in abstract “However, the successful implementation of these recommendations may be challenged by the availability of the state's healthcare resources, necessitating tailored strategies with contextual adaptations.”

Section: Introduction

Comment 4: Are there any particular ethnic groups in India that report a higher incidence of psychological distress such as depression or anxiety as the results of delayed conception?

Response 4: Thank you for the query. Yes, there are certain articles highlighting the higher incidence of mental health issues among marginalized communities. We have added few sentences in the introduction “ “Additionally, certain ethnic groups in LMICs, such as those from marginalized communities or lower socio-economic backgrounds, may report higher incidences of psychological distress due to compounded social pressures and limited access to healthcare.(Selvaraj et al. 2014; Ghia and Rambhad 2023) Studies have shown that women from Scheduled Castes and Scheduled Tribes often face greater challenges in accessing reproductive health services, which can exacerbate mental health issues in cases of delayed conception”

Comment 5: What are the differences between depression and anxiety on one side and psychological distress on the other, given that anxiety and depression are the main causes of psychological distress? This needs more explanation and definitions within the context of the literature.

Response 5: Thank you for the comment and suggestion. There is difference between psychological distress, anxiety, and depression. Hence, for better clarity ( as per your suggestion), we have added a few sentences in the introduction section “While anxiety and depression are specific mental health disorders characterized by distinct diagnostic criteria, psychological distress is a more general term that encompasses a range of emotional disturbances. (Aline Drapeau 2012)Anxiety and depression are often primary contributors to psychological distress, but the latter can also include stress, worry, and other forms of mental discomfort that may not necessarily fit into a specific psychiatric category.(Mirowsky and Ross 2002; Horwitz 2007)”

Section: Methods

Comment 6:.What was the sample size calculation?

Response 6: Thank you for the feedback on sample size calculation. We have included the sentence in the methodology “The minimum sample size needed to estimate infertility with 2% precision and 95% CI from this sample would be 1223, assuming a prevalence of 17% as reported in the literature, and a finite population of 12,500.(Rutstein 2016; Adamson et al. 2011) An allowance of 25% for rejection/ incomplete non-responses was made and the final sample size was 1530.” 

Comment 7: What were the exclusion criteria? Were women who were diagnosed with anxiety and depression, either with or without treatment, excluded? How were they identified?

Response 7: Thank you for the comment. In this study, the population consisted of women who had completed 18 months of follow-up in the WINGS trial without achieving pregnancy. We excluded women who were over 49 years of age or were currently pregnant and we added a sentence in the manuscript “Any women who were > 49 years and currently pregnant were not considered eligible for this study.”

In the primary WINGS trial, all enrolled women underwent mental health screening using the Patient Health Questionnaire (PHQ-9). Women with PHQ-9 scores above 10 were referred to study psychologists, while those with scores above 14 were referred to a psychiatrist at the collaborating study hospital.

In the current study, a cross-sectional survey aimed at assessing the community-level burden of mental health issues, we did not perform any clinical tests prior to recruitment. Our focus was on capturing the mental health status of these women with delayed pregnancy within their community context.

Comment 8:: How was the face-to-face response during data collection eliminated or at least minimized to prevent personal bias? Did the authors address these issues?

Response 8: Thank you for the query. Yes, we tried to minimize the personal bias. For clarity we have added a few sentences in the methodology “To minimize personal bias during face-to-face data collection, interviewers were trained to use neutral language and standardized protocols. Additionally, privacy was ensured during interviews, and participants were reassured about the confidentiality of their responses to reduce social desirability bias” 

Comment 9:: Please provide details on how the women were recruited, even though this study is part of a larger referenced study.

Response 9: Thank you for the suggestion. The recruitment process for this study was carefully designed to ensure a systematic and unbiased selection of participants. Although this study is part of a larger referenced study (WINGS), the recruitment for the current study was conducted separately and followed a distinct protocol.

Participants were recruited from women who had completed 18 months of follow-up in the WINGS trial without achieving pregnancy. Consent to participate in this new study on failure to conceive was sought at the time of their exit from WINGS. These research assistants contacted the women at least 14 days after their exit from WINGS (primary trial) to introduce the new study. Women were recruited consecutively based on their exit dates from WINGS, ensuring that those who met the study criteria (failure to conceive after 18 months) were invited to participate. The sample size for this study was calculated to ensure adequate precision, with a target of 1,530 participants. Recruitment continued until this sample size was achieved, based on preliminary data suggesting that approximately 50% of the WINGS participants had not conceived by the end of the 18 months. This methodical approach ensured a robust and representative sample for assessing the mental health outcomes of interest.

We have added this strategy to the manuscript’s methods section “The recruitment for this study was conducted separately from the WINGS trial, targeting women who had completed 18 months of follow-up without achieving pregnancy. Consent was obtained at the time of their exit from WINGS, and research assistants contacted these women at least 14 days later to introduce the new study. Participants were consecutively recruited based on their exit dates, ensuring those who met the criteria were included. The study aimed for a sample size of 1,530, with recruitment continuing until this target was reached, ensuring a representative sample for assessing mental health outcomes.”

Comment 10: What were the reliability measures (Cronbach's alpha) for each instrument used to collect data?

Response 10: Thank you for the query. Studies from various parts of the world have assessed the reliability of this study instrument i.e., PHQ-4, PHQ-2, and GAD-2 scales, consistently finding good internal consistency. For clarity, we have added this information, along with references, in the methodology section under 'Survey Instrument.“Numerous studies across various settings have demonstrated the PHQ-4 scale’s as well as PHQ-2 scales’ good internal consistency, reporting a Cronbach’s alpha of 0.80..(Kroenke et al. 2009; Manea, Gilbody, and McMillan 2012) Similarly, the GAD-4 (

---

## [Decision Letter · Decision Letter 1]

25 Nov 2024

Prevalence and correlates of symptoms of depression, anxiety,and psychological distress among women of reproductive age with delayed conception in urban and peri-urban low to mid-socioeconomic neighborhoods of Delhi, India: A cross-sectional study.

PONE-D-24-11366R1

Dear Dr. Mazumder,

We’re pleased to inform you that your manuscript has been judged scientifically suitable for publication and will be formally accepted for publication once it meets all outstanding technical requirements.

Kind regards,

Yuan-Pang Wang, M.D., Ph.D.

Academic Editor

PLOS ONE

Additional Editor Comments (optional):

The manuscript is much improved. Authors have addressed all reviewers' concerns and changes were made in the manuscript to reflect their responses. This article is well written and adds new insights for reproductive-age women in India.

Reviewers' comments:

Reviewer's Responses to Questions

**Comments to the Author**

1. If the authors have adequately addressed your comments raised in a previous round of review and you feel that this manuscript is now acceptable for publication, you may indicate that here to bypass the “Comments to the Author” section, enter your conflict of interest statement in the “Confidential to Editor” section, and submit your "Accept" recommendation.

Reviewer #3: All comments have been addressed

2. Is the manuscript technically sound, and do the data support the conclusions?

Reviewer #3: Yes

3. Has the statistical analysis been performed appropriately and rigorously? 

Reviewer #3: Yes

4. Have the authors made all data underlying the findings in their manuscript fully available?

Reviewer #3: Yes

5. Is the manuscript presented in an intelligible fashion and written in standard English?

Reviewer #3: Yes

6. Review Comments to the Author

Reviewer #3: I am willing to accept your paper,

“Prevalence and correlates of symptoms of depression, anxiety, and psychological distress among women of reproductive age with delayed conception in urban and peri urban low to mid-socioeconomic neighborhoods of Delhi, India: A cross-sectional- study”

The author has very clearly explained all the queries raised by the reviewers. The author also added the latest references highlighted by the reviewers.

Hence, I am ready to accept this paper

7. PLOS authors have the option to publish the peer review history of their article (what does this mean?). If published, this will include your full peer review and any attached files.

Reviewer #3: **Yes: **Dr.T.Beena

---

## [Editor Report · Acceptance letter]

15 Dec 2024

PONE-D-24-11366R1 

PLOS ONE

Dear Dr. Mazumder, 

I'm pleased to inform you that your manuscript has been deemed suitable for publication in PLOS ONE. Congratulations! Your manuscript is now being handed over to our production team.

Kind regards, 

on behalf of

Dr. Yuan-Pang Wang 

Academic Editor

PLOS ONE